# OpenReview forum: "Context Steering: Controllable Personalization at Inference Time"
_ICLR.cc/2025/Conference — ICLR 2025 Poster_

### Official Review · Reviewer_J3fF · 2024-11-02

**Soundness:** 3
**Presentation:** 2
**Contribution:** 3
**Rating:** 6
**Confidence:** 4

**Summary:**

This paper introduces the CoS method for controlling the personalization of LLM generation results during inference. CoS operates by calculating the difference between LLM outputs with and without personalized context, and subsequently incorporating this difference into the original outputs with a weight parameter, lambda, to adjust the level of personalization. A higher lambda corresponds to a greater degree of personalization. The core idea shares similarities with existing counterfactual methods; however, applying it to control personalization is novel. Besides proposing CoS, the paper presents a method for inferring lambda in reverse from a given generation result, aiding in the identification of implicit intents, such as the 'degree of hate in statements.'

**Strengths:**

S1. Controlling the level of personalization by using the difference between LLM outputs with and without personalized context appears reasonable and straightforward, with the entire process completed at inference time.
S2. The approach of inferring implicit intents from a given generation result is interesting.
S3. A variety of experiments are presented.

**Weaknesses:**

W1. The experimental evaluation appears insufficiently convincing. It would be beneficial to include more evaluations with objective metrics. For instance, incorporating experiments conducted on established benchmarks for LLM personalization [1] and recommendation [2] would strengthen the analysis.

W2. Some experiments and their results are difficult to follow, such as those related to movie recommendations and hate identification. In the recommendation experiments, it is unclear how the baselines—multi-turn Q&A and in-context learning—are compared under different lambda values. Moreover, the results indicate a higher win rate for these baselines. How do these outcomes demonstrate the proposed method's advantages? For the hate identification experiments, the results are not presented in a clear manner.

W3. The method's effectiveness seems dependent on the LLM’s existing ability to generate personalized responses for a given context. This suggests that the approach amplifies current personalization rather than fundamentally enhancing it. For example, if an LLM's personalization is flawed, the method cannot correct it. This limitation indicates that the approach may not serve as a replacement for traditional tuning-based methods.

W4. The advantages of this method over prompt-based approaches (e.g., the multi-turn Q&A baseline) or in-context learning are not clearly outlined.

W5. Table 2 does not include results for lambda=0. Providing these results would offer a more comprehensive view of the evaluation.

[1] LaMP: When Large Language Models Meet Personalization.
[2] BARS: Towards Open Benchmarking for Recommender Systems.

**Questions:**

The main concerns have been outlined under Weaknesses. Below are some additional questions:

Q1. When adding the difference to the original model's predictions, how do you ensure that the generated results remain fluent, coherent, and meaningful?

Q2. Could you provide an example illustrating how to compute lambda and the degree of hate using equations (4) and (5)?

---

> ### Author Response · Authors · 2024-11-27
> **Evaluated on LaMP; added clarification on multi-turn Q&A, ICL, hate classification; added lambda=0 for Table 2.**
>
> > Include evaluations on established benchmarks
>
> We thank the reviewer for the suggestions. Upon careful review, we think that there are some gaps in applying CoS on recommender systems in [2]. We have followed the procedures in [1] and applied CoS on LaMP-7 for tweet paraphrasing (the other tasks exceed our current context length limit). Our results are included in Appendix C.5, where we use the training set to find a set of high-performing lambda values and evaluate them on the held-out set. Interestingly, we find that lambda=-0.2 and lambda=-0.5 achieve the overall best performance, outperforming the proposed prompt (equivalent to lambda=0) in LaMP. We believe that the proposed prompt is highly verbose, and using smaller lambdas (i.e. lambda=-0.5) reduces this redundancy, leading to better generation. We think that this shows CoS’s utility in controllably setting the influence of the context.
>
> > How multi-turn Q&A and in-context learning are compared under different lambda values. How do these outcomes demonstrate the proposed method's advantages?
>
> We have updated Figure 4 to highlight that the multi-turn Q&A and in-context learning are generated under 0,1,2 and 3 rounds of interactions (ICL using the demonstration from GPT-4). The lambda values for CoS are selected from held-out examples such that the outputs roughly match the demonstration from GPT-4. Note that while we only plot lambda=0, 1, 1.5, and 3, CoS can provide more fine-grained results by setting lambda values to lie in between these values.
> Compared to ICL: we find that ICL fails to generate fine-grained personalizations.
> Compared to multi-turn QA: multi-turn QA leads to higher computational cost after round = 2, and unlike CoS, multi-turn QA cannot interpolate between discrete rounds.
>
> > For the hate identification experiments, the results are not presented in a clear manner. How to compute lambda using equations (4) and (5)?
>
> We have updated the descriptions for classifying and quantifying implicit hate and added formula for how we computed the most likely context and inferred the most likely lambda. Concretely to infer lambda, we perform forward inference over a set of increasing lambda spanning from -1 to 3 (interval 0.1), and select the lambda with the highest forward probability given the context (hateful intent) and generation (tweet).
>
> > The method's effectiveness seems dependent on the LLM’s existing ability.
>
> We fully agree with the author that CoS merely serves as a steering technique and the main generation capability comes from the base language model. We do find that for relevant contexts, LLM is highly steerable and for irrelevant contexts (appendix C.2), the generation is not influenced. There are limitations such as CoS cannot steer base text model towards instruction following, which can be achieved by tuning based methods.
>
> > Table 2 does not include results for lambda=0.
>
> We have updated Table 2 to include generation for lambda = 0.
>
> > Ensure that the generated results remain fluent, coherent, and meaningful.
>
> To ensure coherence, as well as ensuring that the generation is diverse and not repetitive, we added an additional study in sec 4.1 and appendix C.4 where we measure coherence and diversity of generated texts following Li et al 2023. We find that CoS naturally leads to diverse and coherent generations and that the lambda value has a small influence on both metrics.

---

### Official Review · Reviewer_WxW1 · 2024-11-03

**Soundness:** 3
**Presentation:** 3
**Contribution:** 3
**Rating:** 6
**Confidence:** 4

**Summary:**

The paper introduces Context Steering (CoS), a method to personalize large language model (LLM) outputs at inference time. This is done by providing the user's characteristics and preference as context, and adjusting the influence of provided context using a contextual influence function. The influence of this function on the token probabilities can be adjusted, to control how personalized the output is to the given context. Applications of CoS include personalized recommendations involving topics such as movies, travel, cooking, etc. Besides this, the paper also introduces a Bayesian inference model by inverting the CoS probability model. This is used for classifying and quantifying implicit hate speech. Further applications of this Bayesian model include identifying tones in open-ended statements and online content moderation.

**Strengths:**

1. CoS is a simple method of personalizing LLM outputs to context, without requiring fine-tuning, or prompt tuning. The method saves on the cost and effort needed for training or prompt-tuning, while being effective in the tests carried out by the authors.
2. The framework can be used directly across many personalization contexts. Fine-tuning or prompt-tuning would require re-tuning for each new context.
3. The experiments show promise, and include human evaluations, GPT4 evaluations, and comparisons with baseline models, across various personalization contexts and implicit hate settings.

**Weaknesses:**

1. Limited contexts: While CoS is effective for single, straightforward contexts (e.g., "I like {genre}"), user preferences are often more complex, involving various (possibly conflicting) likes and dislikes.  It would be interesting to see the method's performance under more sophisticated and detailed contexts.
2. The baseline experiments in Figure 4 are unclear to me. How are various values of lambda used in the case of in-context learning and multi-turn QA? Also, could the supposedly worse performance of ICL be fixed via prompt tuning?

**Questions:**

See weaknesses.

---

> ### Author Response · Authors · 2024-11-27
> **Added results on conflicting multi-contexts; added clarification and discussion on CoS vs ICL.**
>
> > Complex and conflicting likes and dislikes in user preferences.
>
> We thank the reviewer for the suggestions. For multiple and conflicting contexts, we find that CoS scales well to this setting. We have added additional results in Appendix B, qualitatively showing how CoS can be applied to multiple contexts, both non-conflicting and conflicting contexts. Interestingly, in both cases, we can reliably control the influence of each context by its lambda. For conflicting contexts, the generation acknowledges both directions and implicitly leans towards one depending on the influence strength (lambda).
>
> We acknowledge that our study is limited to settings where the context and the prompt are cleanly separable. In general, they may be entangled in the same text. While we believe that one can define a soft context mask and apply the steering technique, we leave such investigation to future works.
>
> > Figure 4: How are various lambda values used in the case of in-context learning and multi-turn QA? Prompt tuning for ICL?
>
> We have updated Figure 4 and the paper for more detailed descriptions. The results for multi-turn Q&A and in-context learning are generated under 0,1,2 and 3 rounds of interactions (ICL using the demonstration from GPT-4). The lambda values for CoS are selected from held-out examples such that the outputs roughly match the demonstrations of GPT-4. Note that while we only plot lambda=0, 1, 1.5, and 3, CoS can provide more fine-grained results by setting lambda values to lie in between these values.
>
> As noted in Figure 4, ICL has a poor ability to provide fine-grained personalizations. While we believe that prompt-tuning has the potential to improve ICL generations, it is also challenging to compute the objective/loss for prompt-tuning, since “the level of personalization” is a rather subjective measurement. Because of this, we think that CoS is a simple and direct method for controllable personalization.

---

### Official Review · Reviewer_jnRg · 2024-11-04

**Soundness:** 4
**Presentation:** 4
**Contribution:** 3
**Rating:** 8
**Confidence:** 3

**Summary:**

This paper introduces Context Steering (CoS), a method for controlling the influence of context in Large Language Model generated text. The key idea behind CoS is to quantify the impact of context by comparing the output probabilities of the LLM with and without the given context. This key parameter lambda allows CoS to adjust the level of contextual influence on the generated text.

The paper demonstrates the effectiveness of CoS in various applications. One application is generating personalized recommendations, where CoS can tailor the LLM's output to specific user preferences. Another application is inferring relationships between open-ended texts, which can be used for tasks like classification and quantification of implied statements.

**Strengths:**

-The paper is well written and easy to read.

-The proposed approach to achieve personalization is simple, novel, and training-free, applicable to various LLMs.

-Extensive experiments demonstrate strong performance in personalized recommendations, identification of implicit intents and quantification of extent of “personalization”.

-The experimental analysis is comprehensive.

**Weaknesses:**

-Focused primarily on a single context. The paper primarily focuses on scenarios with a single, dominant context. However, real-world situations often involve multiple, potentially conflicting contexts. For example, in the movie case, the user might be interested in comedy movies, science fiction but also movies with great storytelling.

-Limited Discussion on Computational Complexity: While the authors mention that CoS requires twice the amount of compute compared to a vanilla forward pass, they do not provide a detailed analysis of its computational complexity. A more in-depth analysis of how the computational cost scales with input length, context size, and lambda values would be beneficial.

-Limited discussion on the impact of CoS on other tasks such as reasoning and creativity.

**Questions:**

-How can CoS be extended to handle multiple contexts with varying levels of influence? How would the method resolve potential conflicts between different contexts?

-Can you provide a more comprehensive analysis of the computational complexity of CoS? How does the computational cost vary with different parameters and input characteristics (input length, context size, and lambda values)?

-How does the CoS approach affect the LLM's ability for other tasks, e.g., reasoning and creativity? It might be worth some discussion here.

-The appendix seems missing from the manuscript. Is it accidentally omitted?

---

> ### Author Response · Authors · 2024-11-27
> **Multiple context validated, added discussions and experiments on computational cost, reasoning and creativity**
>
> > Extend to handle multiple contexts with varying levels of influence?
>
> Indeed, we find that CoS scales very well to multiple contexts. We have added additional results in Appendix B, qualitatively showing both non-conflicting and conflicting contexts. Interestingly, we find that in both cases, we can reliably control the influence of each context by its lambda. For conflicting contexts, the generation acknowledges both directions, and implicit leans towards one depending on the influence strength (lambda).
>
> > Needs analysis of computational cost
>
> We have added an analysis of the computational cost in section 4.2. More specifically, CoS scales linearly with the number of contexts, quadratically with the total sequence length (max context + prompt + generation), and linearly with candidate set (number of candidate lambdas).
>
> > How does the CoS approach affect the LLM's ability for other tasks, e.g., reasoning and creativity?
>
> For reasoning tasks, we provide additional study in section 4.1 and appendix C.2 based on OpenbookQA. We find that with relevant contexts, small lambda does not influence the ability, and large lambda leads to small decrease. We also find that irrelevant falsehoods as contexts do not influence reasoning.
> For creativity, we provide additional study in section 4.1 and appendix C.4 where we measure coherence and diversity of generated texts following Li et al 2023. We find that lambda has little influence on both metrics.
>
> > The appendix seems missing from the manuscript. Is it accidentally omitted?
>
> We apologize for the mistake. The original appendix was included in the supplementary material. We have included it in the updated manuscript.

---

### Meta-Review · Area_Chair_1qmT · 2024-12-21

**Metareview:**

In this paper, the authors propose an approach to control the impact of context in text generation and demonstrate its effectiveness in applications such as personalized recommendations and inferring relationships between open-ended texts. Reviewers found the paper well-written, the approach straightforward to implement, and the experiments thorough. There was some discussion about whether the approach could handle complex contexts, as well as limited exploration of computational complexity and applicability to additional tasks. Overall, I believe this paper exceeds the acceptance threshold.

**Additional Comments On Reviewer Discussion:**

The authors provided additional experiments and clarified questions regarding the experimental results. Although the reviewers did not respond further, I believe the authors have adequately addressed all concerns.

---

### Decision · Program_Chairs · 2025-01-22

Accept (Poster)